# Can serum interleukin 34 levels be used as an indicator for the prediction and prognosis of COVID-19?

Doğu Karahan[1]*, Hasan Ata Bolayir[2], Aslı Bolayir[3], Bilgehan Demir[4], Önder Otlu[5], Mehmet Erdem[5]

1 Department of Internal Medicine, Malatya Turgut Özal University School of Medicine, Malatya, Turkey,
2 Department of Cardiology, Malatya Turgut Özal University School of Medicine, Malatya, Turkey,
3 Department of Neurology, Malatya Turgut Özal University School of Medicine, Malatya, Turkey,
4 Department of Emergency Medicine, Malatya Turgut Özal University School of Medicine, Malatya, Turkey,
5 Department of Medical Biochemistry, Malatya Turgut Özal University School of Medicine, Malatya, Turkey

* dogu.karahan@ozal.edu.tr

## Abstract

### Objective

Interleukin 34 (IL-34) is a molecule whose expression is increased in conditions such as autoimmune disorders, inflammation, and infections. Our study aims to determine the role of IL-34 in the diagnosis, follow-up, and prognosis of Coronavirus Disease-19 (COVID-19).

### Method

A total of 80 cases were included in the study as 40 COVID-19 positive patient groups and 40 COVID-19 negative control groups. The COVID-19-positive group consisted of 20 intensive-care unit (ICU) patients and 20 outpatients. Serum IL-34, c-reactive protein (CRP), ferritin, D-dimer, troponin I, hemogram, and biochemical parameters of the cases were studied and compared between groups.

### Results

IL-34 levels were significantly higher in the COVID-19-positive group than in the negative group. IL-34 levels increased in correlation with CRP in predicting the diagnosis of COVID-19. IL-34 levels higher than 31.75 pg/m predicted a diagnosis of COVID-19. IL-34 levels did not differ between the outpatient and ICU groups in COVID-19-positive patients. IL-34 levels were also not different between those with and without lung involvement.

### Conclusion

While IL-34 levels increased in COVID-19-positive patients and were successful in predicting the diagnosis of COVID-19, it was not found to be significant in determining lung involvement, risk of intensive care hospitalization, and prognosis. The role of IL-34 in COVID-19 deserves further evaluation.

**Data Availability Statement:** The data set has been added as a supplementary file along with the revision files.

**Funding:** The author(s) received no specific funding for this work.

**Competing interests:** The authors have declared that no competing interests exist.

## Introduction

Coronaviruses are enveloped, single-stranded RNA viruses with ~30 kb of genetic material. SARS-CoV-2 has been classified as a beta coronavirus subtype [1]. Symptoms of patients infected with SARS-CoV-2 range from minimal to severe respiratory failure (SARS) [1]. The infection that developed due to SARS-CoV-2 and caused the pandemic was named COVID-19. Pneumonia and lung damage often develop in moderate to severe manifestations of COVID-19 [2]. It has been suggested that the variation in symptom severity and the negative outcomes in patients progressing to multi-organ failure are due to differences in the immunological profile of the host, common thrombo-embolic events in the whole body, and hypoxia rather than virus variation [3, 4]. Viral infection is often accompanied by an excessive inflammatory response called "cytokine storm", which damages endothelial integrity and facilitates endothelial induction [5]. Again, thrombo-embolic conditions developing in many organs, especially pulmonary thrombo-embolism, deep venous thrombosis, and acute cardiac and brain stroke, are associated with multi-organ failure and increased risk of death [4]. The immunopathology of COVID-19 is currently at the center of basic and clinical research. Multiple studies have reported extremely high levels of pro-inflammatory cytokines in COVID-19. However, we still have limited information about the mechanisms of cytokine storm, thrombo-embolism, and hypoxia in severe COVID-19.

Interleukin-34 (IL-34) is a cytokine that was identified as the tissue-specific ligand of the CSF-1 receptor (CSF-1R) in a comprehensive proteomic analysis in 2008 [6, 7]. Structurally, IL-34 belongs to the short-chain helix family of hematopoietic cytokines but does not show a clear consensus structural domain, motif, or sequence homology with other cytokines. IL-34 is synthesized as a secreted homodimeric glycoprotein that binds to the extracellular domains of CSF-1R and receptor-type protein-tyrosine phosphatase-zeta (PTP-$\zeta$) in addition to the chondroitin sulphate chains of syndecan-1 [7, 8]. IL-34 expression can be induced by various stimuli such as DNA-damaging agents, chemical stressors, pro-inflammatory cytokines, viral infection, and vitamin D. In humans, IL-34 is found in a variety of tissues, including heart, brain, lung, liver, kidney, prostate, and colon. At the cellular level, IL-34-producing cells include immune cells, epithelial cells, adipocytes, and cancer cells. In addition to cellular adhesion and migration, IL-34 is involved in the activation of several signalling pathways that regulate major cellular functions, including proliferation, differentiation, survival, metabolism, and cytokine/chemokine expression [7, 9].

In pathological conditions, increasing or decreasing changes in IL-34 expression play a role in the pathogenesis of diseases and are associated with progression, severity, and chronicity [10]. Indeed, several studies have confirmed that IL-34 expression is increased at mRNA and protein levels in the context of various diseases, including autoimmune disorders, inflammation, and infections [11–14].

In our study, we aimed to determine the role of IL-34 in the specific case of COVID-19, based on the roles of IL-34 in inflammatory, cytokine, and immune response, which have been identified in previous studies. We wanted to evaluate the possible relationship between IL-34 and COVID-19 disease severity, SARS occurrence, and prognosis. Again, we planned to compare IL-34 with the other parameters used in the diagnosis, follow-up, and prognosis evaluation of COVID-19.

## Material and methods

This prospective study was carried out in Malatya Turgut Özal University Faculty of Medicine Training and Research Hospital COVID-19 intensive care unit and COVID-19 clinic. Ethical approval for this study was obtained from the local ethics committee of Malatya Turgut Özal

University with the number 2022–40. The study was conducted according to the Declaration of Helsinki. The study was carried out on 80 cases. Cases were included in the study between the period of 01.09.2022 and 24.10.2022, after the ethics committee approval. Forty cases were determined as the COVID-19 positive patient group and 40 cases as the COVID-19 negative control group. While 20 of the COVID-19-positive group were inpatients in the ICU, 20 of them consisted of patients evaluated in the outpatient COVID-19 clinic. Written informed consent was obtained from the outpatients included in the study and from the relatives of the uncooperative and disoriented patients in the ICU. The collected data were accessed between 04.11.2022 and 01.12.2022 for research purposes.

The diagnosis of COVID-19 was confirmed by the reverse transcriptase PCR (RT-PCR) method, by taking nasopharyngeal swabs from patients with symptoms. Among the participants included in the study, those with positive COVID-19 RT-PCR results were included in the patient group, and those with negative results were included in the control group. The COVID-19 negative group consisted of cases with a similar age and demographic structure as the COVID-19 positive group, who applied with complaints and symptoms of respiratory tract infection, and whose PCR test result was negative. The ICU group consisted of cases with widespread lung involvement, hypoxic, requiring high dose oxygen from a reservoir, or requiring non-invasive and invasive mechanical ventilation. The three COVID-19 positive patients in the ICU without lung involvement were hemodynamically unstable, hypoxic and hypotensive. The patients in the ICU were receiving antibiotherapy, mucolytic, enoxaparin, dexamethasone and supportive treatments. The history of the cases was questioned, and chronic diseases and smoking histories were recorded. The auto-immune diseases and cancer stories of the cases were questioned and evaluated, and cases with such disease clinics and stories were not included in the study. Blood samples were taken from the patients in the ICU group before the first dexamethasone administration. Our exclusion criteria in the study were dexamethasone administration, having a diagnosis of cancer and auto-immune disease, and being under the age of <18 years.

IL-34 serum concentrations were determined simultaneously in the first 24 hours in cases admitted to the ICU or COVID-19 clinic. Serum IL-34 levels were determined by the same method from serum samples taken from control group cases without a diagnosis of COVID-19, by the same researcher.

Blood samples for IL-34 were taken from the individuals into a gel separator (serum) tube in the morning after overnight fasting. After the blood collection, the serums were kept at room temperature for 30 minutes, coagulated, and then centrifuged for 10 minutes. After centrifugation, serum samples were separated from the tubes and transferred to sterile 1.5 mL Eppendorf tubes. Serum samples were kept at -80˚C until biochemical analysis.

## Interleukin-34 levels measurement

Serum samples stored at -80˚C were prepared for the experiment by providing appropriate conditions. Interleukin-34 (Bioassay Technology Laboratory, Cat. No: E0043Hu, China) levels were studied by using Enzyme-Linked ImmunoSorbent Assay (ELISA) kits. The unit for IL-34 was pg/mL.

SARS-CoV-2 PCR tests of the samples obtained from the nasopharyngeal swabs of the cases were studied on the Rotor-Gene Q Real Time PCR (QIAGEN, Hilden Germany) device.

Blood samples were taken from all patients for necessary laboratory investigations. For clinical evaluation of cases, hemogram, glucose, total cholesterol, low-density lipoprotein cholesterol (LDL), creatinine, aspartate transaminase (AST), albumin, c-reactive protein (CRP), D-dimer, troponin I (TnI), and ferritin tests were studied. Blood samples were taken from the

antecubital vein under sterilization conditions and put into appropriate tubes for necessary examinations.

The complete blood count of the patients was automatically studied in the Sysmex Corporation (XN-10, Kobe, Japan) device in Malatya Turgut Özal University Training and Research Hospital Hematology Laboratory from 2 ccs of blood collected in standard EDTA tubes.

For biochemical analyses, blood samples taken without anticoagulant were incubated at 37°C for 20 minutes and then centrifuged at 1800 rpm for 10 minutes, and serum samples were obtained. CRP, glucose, total cholesterol, LDL, creatinine, albumin, AST, ferritin, and D-dimer levels were automatically studied in Abbott Architect c16000, (Illinois, United States of America) autoanalyzer with Abbott brand (Sentinel Diagnostic Illinois, United States of America) commercial kits. The unit for CRP, glucose, total cholesterol, LDL, and creatinine was mg/dL, the unit for AST was U/L, and the unit for albumin was g/dL.

Troponin I level was studied with Roche Diagnostics (Cobas E601, Tokyo, Japan) device. The unit for ferritin, D-dimer, and TnI was ng/mL.

The cases were also evaluated and compared in terms of COVID-19 lung involvement. The lung involvement of the cases was evaluated according to the COVID-19 Reporting and Data System (CO-RADS) [15] with thorax computed tomography (CT) and chest radiography findings. According to this classification, those evaluated as CO-RADS 4–5 were accepted as COVID-19 lung involvement. Those classified as CO-RADS 1 and 2 were considered negative for COVID-19 lung involvement. Cases classified as CO-RADS 3, where a clear distinction could not be made between other diseases causing lung involvement and COVID-19 involvement, were not included in the study. Thorax CT scans were performed on a Siemens 128-section device.

## Statistical analysis

Statistical analyses were conducted using the Statistical Package for the Social Sciences for Windows 21.0 (SPSS Inc. Chicago, IL, USA). Where appropriate, the Kolmogorov-Smirnov test, $x^2$ test, one-way analysis of variance (ANOVA), or Kruskal-Wallis test was used for the comparison of the groups. Normally distributed variables were shown as mean and standard deviation, while non-normally distributed variables were shown as median (minimum-maximum) values. Descriptive statistics are given as a percentage and absolute values. Differences between subgroups were revealed using Dunn's procedure. Data were analysed to identify whether IL-34 was independently associated with the risk of COVID-19 by using univariate logistic and multivariate logistic regression models. Univariate analyses considered the following variables: White blood cell (WBC), CRP, age, sex, smoking, albumin, D-dimer, TnI, and ferritin. Covariates with $p < 0.1$ from univariate logistic regression were included for multivariate analysis. We performed receiver operating characteristic (ROC) analysis to find the most sensitive IL-34 cut-off level for identifying patients with COVID-19. $p < 0.05$ was accepted to be statistically significant. The sample size of this study was determined by power analysis. According to the calculation made using the G*power 3.1 program; the sample size was determined to be at least 80 (40 in each group) with an effect size of 0.75, margin of error of 0.05, confidence level of 0.95, and representative power of the universe of 0.95 [16].

## Results

Demographic and laboratory characteristics of the study population are shown in Table 1. Of the cases included in the study, 42 were male and 38 were female. Demographic parameters, complete blood count, lipid profile, AST, and creatinine were not different between the groups ($p > 0.05$). IL-34, CRP, D-dimer, TnI, and ferritin levels were statistically significantly higher;

**Table 1. Baseline characteristics and laboratory parameters of the study population.**

|  | Control group (n = 40) | COVID-19+ ICU group (n = 20) | COVID-19+ outpatient group (n = 20) | p-Value |
|---|---|---|---|---|
| Age (years) | 71.7 (62–88) | 78.15 (67–90) | 71.9 (58–83) | 0.13 |
| Men, n (%) | 20 (50) | 12 (60) | 10 (50) | 0.31 |
| Hypertension, n (%) | 18 (45) | 10 (50) | 9 (45) | 0.09 |
| Diabetes mellitus, n (%) | 16 (40) | 9 (45) | 9 (45) | 0.19 |
| Smoking, n (%) | 12 (30) | 8 (40) | 10 (50) | 0.07 |
| Total cholesterol (mg/dL) | 185 (144–221) | 192 (111–254) | 188 (144–222) | 0.47 |
| LDL (mg/dL) | 93 (69–131) | 102 (75–119) | 96 (86–111) | 0.09 |
| Creatinine (mg/dL) | 0.79 (0.49–1.12) | 1.02 (0.46–1.77) | 0.76 (0.53–1.16) | 0.42 |
| AST (U/L) | 22 (12–40) | 29 (15–53) | 23 (15–30) | 0.12 |
| Haemoglobin (g/dL) | 14.2 (10.5–16.1) | 12.12 (10.2–15.7) | 14.5 (10.5–20.7) | 0.14 |
| WBC ($10^3$/mm$^3$) | 10.04 (5.33–16.81) | 14.5 (5.03–27.61) | 12.5 (7.27–19.44) | 0.06 |
| *Albumin (g/dL)* | *4.2 (3.5–5.0)* | *3.1 (2.3–4.1)* | *3.8 (2.4–4.7)* | *0.04* |
| Platelet Count ($10^6$/mL) | 297 (202–438) | 248 (63–425) | 261 (153–370) | 0.76 |
| *D-dimer (ng/mL)* | *0.98 (0.69–1.22)* | *1.42 (0.92–2.1)* | *1.04 (0.85–1.27)* | *0.04* |
| *Troponin I (ng/mL)* | *0.1* | *0.36 (0.16–1.54)* | *0.18 (0.16–0.26)* | *0.04* |
| *Ferritin (ng/mL)* | *24.4 (3.5–52)* | *164 (59.4–443)* | *34 (20–53)* | *0.002* |
| *CRP (mg/dL)* | *1.33 (0.84–1.94)* | *7.24 (1.69–15.13)* | *5.66 (1.38–14.1)* | *0.019* |
| *IL-34 (pg/mL)* | *16.1 (10.03–26.5)* | *38.6 (20.51–51.7)* | *29.5 (12.44–52.5)* | *0.001* |

Bold values are used to demonstrate the statistical significance. All values are presented as mean and min-max, or n (%). Abbreviations: LDL: low-density lipoprotein cholesterol, AST: aspartate transaminase, WBC, white blood cell, CRP: c-reactive protein, IL-34: Interleukin 34

albumin levels were statistically significantly lower in the COVID-19-positive group compared to the control group (p<0.05).

In a pairwise comparison, while the COVID-19-positive group had statistically significantly higher IL-34 levels compared to the control group (p<0.05), there was no statistically significant difference between the IL-34 levels of the COVID-19-positive groups (ICU or outpatient) (p > 0.05) (Fig 1). In a pairwise comparison, CRP levels were statistically significantly higher in the COVID-19-positive group than in the control group (p<0.05), but no statistically significant differences were found between COVID-19-positive groups (ICU or outpatient)

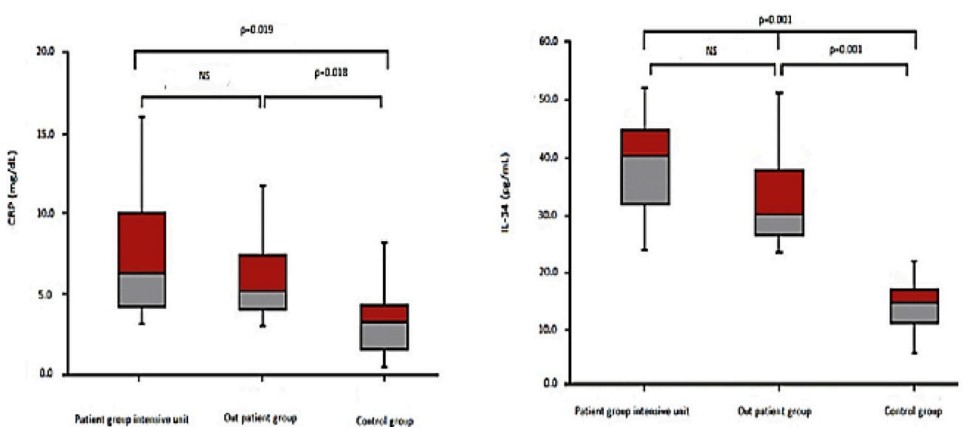

**Fig 1. Comparison of CRP and IL-34 levels between COVID-19-positive groups (ICU and outpatient) and control group by pairwise comparison.** CRP: c-reactive protein, IL-34: Interleukin 34, NS: non-significant.

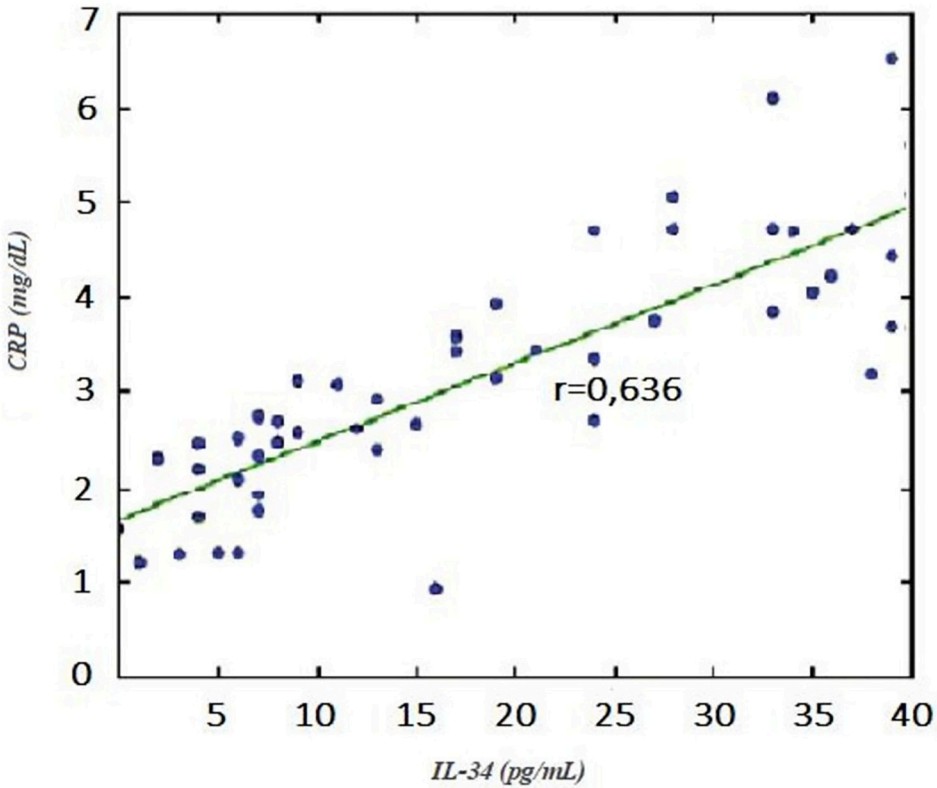

**Fig 2. Correlation analyses of plasma IL-34 and CRP levels in COVID-19-positive patients.** CRP: c-reactive protein, IL-34: Interleukin 34.

(p > 0.05) (Fig 1). In patients with COVID-19, plasma IL-34 levels correlated positively with CRP (r = 0.636, p = 0.029) (Fig 2). But, in patients with COVID-19, plasma IL-34 levels did not correlate with pulmonary infiltration (r = 0.124, p = 0.48). In terms of lung involvement, a significant correlation was found with ferritin (r = 0.726, p = 0.032), D-dimer (r = 0.644, p = 0.044), and TnI (r = 0.704, p = 0.030). The multivariate analysis with adjustment for potential confounding variables revealed that higher IL-34 and CRP levels were independently associated with COVID-19 (Table 2).

ROC analysis showed that IL-34 levels higher than 31.75 pg/mL could predict COVID-19 (p < 0.001) as shown in Fig 3 (sensitivity: 78%; specificity: 69%; area under the curve: 0.734; 95% confidence interval: 0.582–0.820).

## Discussion

The pathophysiology of COVID-19, the underlying inflammatory process, and the risk factors for disease progression have still not been resolved clearly and are interesting topics that continue to be studied. Many inflammatory and immunological markers have been studied in terms of inflammatory process and prognosis in COVID-19. However, only one study was found in the literature review related to the role of IL-34 in COVID-19. According to the results of our study on this subject, IL-34 increased significantly in COVID-19 positivity and predicted the disease, but did not provide clear information in terms of lung involvement, risk of intensive care hospitalization, and prognosis.

**Table 2. Univariate and multivariate logistic regression analysis representing the independent predictors of COVID-19.**

| Variables | Univariate regression analysis | | Multivariate regression analysis | |
|---|---|---|---|---|
| | Exp(β) (95% CI) | p-Value | Exp(β) (95% CI) | p-Value |
| Age | 1.034 (0.975–1.097) | 0.260 | – | – |
| Male sex | 0.561 (0.217–1.449) | 0.233 | – | – |
| Smoking | 1.000 (0.229–4.361) | 1.000 | – | – |
| Ferritin | 0.736 (0.522–1.038) | 0.081 | 0.777 (0.532–1.137) | 0.194 |
| D-dimer | 0.975 (0.940–1.011) | 0.078 | – | – |
| Troponin I | 0.209 (0.016–2.718) | 0.074 | – | – |
| WBC | 1.015 (0.991–1.040) | 0.095 | – | – |
| Albumin | 0.787 (0.608–1.018) | 0.068 | 0.789 (0.589–1.056) | 0.111 |
| *CRP* | *1.221 (1.045–1.426)* | *0.012* | *1.201 (1.005–1.435)* | *0.044* |
| *IL-34* | *1.008 (1.003–1.014)* | *0.004* | *1.007 (1.001–1.013)* | *0.021* |

CI, confidence interval; CRP, C-reactive protein; WBC, white blood cell, IL-34: Interleukin 34

It has been reported that the expression of IL-34 increases in infections, inflammatory conditions, and autoimmune rheumatological diseases [17, 18]. There are studies reporting that it is elevated in autoimmune diseases such as systemic lupus erythematosus, arthritis, systemic

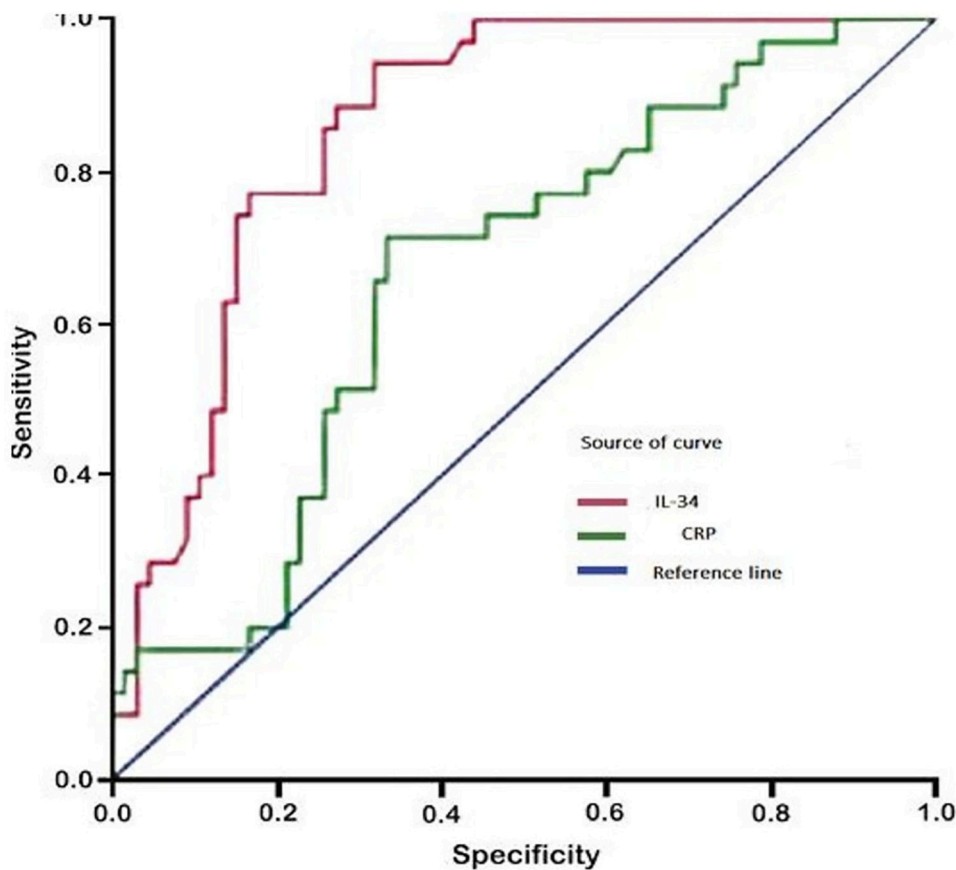

**Fig 3. ROC curve analyses of the predictive power of plasma IL-34 level for COVID-19.** AUC: Area under the curve, CRP: c-reactive protein, CI: confidence interval, ROC: receiver operating characteristic.

sclerosis, and inflammatory bowel diseases [18]. There are a limited number of studies evaluating the relationship between IL-34 and viral infections. Preisser L et al. in their study on hepatitis C virus (HCV) infection, stated that HCV increased the levels of macrophage colony-stimulating factor (M-CSF) and IL-34 [14]. Yu G et al. in their study evaluating the level of IL-34 in influenza A viral infection found that the level of IL-34 increased in serum and peripheral blood mononuclear cells [19]. In our study, significant elevation of IL-34 with CRP in COVID-19 seems to be compatible with previously reported results in terms of elevation of IL-34 in infections and inflammatory processes. As a result of our study, a valuable result has been obtained regarding the use of IL-34 in the evaluation of SARS-Cov-2 and similar infections. In addition, although IL-34 was elevated in COVID-19 infection, levels appeared to be undifferentiated in patients with COVID-19 lung involvement and in ICU patients. Based on this, it can be interpreted that although inflammation develops in COVID-19, non-inflammatory processes may play a more prominent role in the progression of COVID-19 and the worsening of patients' clinical conditions.

It is stated that in addition to SARS, thrombo-embolic events, hypoxia, genetic and demographic variables, and co-morbidities play a role in the impairment of the clinical situation of COVID-19 patients [20–22]. There are hundreds of studies in the literature on COVID-19 and thromboembolic complications, which have been done during the pandemic and indicate the increased risk. Studies on this subject have stated that although coagulation-related complications are observed primarily in the lungs, heart, kidneys, and brain in COVID-19, the frequency of arterial and venous thromboembolic events increases throughout the body. These studies reported that the risk of ICU admission, morbidity, and mortality increased with thromboembolic events [23–25]. Cheng NM et al. in their review study, in which they evaluated 68 publications, reported that the incidence of thromboembolic events increased in severe cases, pulmonary embolism (PE) and deep venous thrombosis (DVT) were the most common forms, and D-dimer was an independent risk factor for thromboembolic events [26]. Many studies reported that increased D-dimer and TnI due to coagulation were associated with the worsening prognosis of COVID-19 and increased mortality [27–30]. Kaufmann CC et al. reported that IL-34 was not associated with cardiovascular disease and early mortality in their study on patients with cardiovascular disease and hospitalized due to COVID-19. They also indicated that short-term mortality increased significantly in those with multiple cardiovascular diseases [31]. In the results of our study, a statistically significant difference was found between the control group, outpatient COVID-19 cases, and ICU patients in terms of D-dimer and TnI levels. This result, together with the above studies, supports that the coagulation tendency is one of the most important factors in worsening the clinical condition of the cases. In our study, the fact that the level of IL-34, an inflammatory and immunity-related molecule, did not differentiate between ICU and outpatients, supports the thesis that the increased susceptibility to coagulation and thrombo-embolism is more prominent in the worsening of the patient's clinical status and the increased risk of death.

The elevation and role of CRP in COVID-19 is a well-studied topic. Colak A et al. stated in their studies that CRP, lymphocyte, and % monocyte were independent risk factors for COVID-19 [32]. Kaftan AN et al. in their study on the evaluation of the diagnostic accuracy before taking RT-PCR test for definitive diagnosis in patients presenting with symptoms suspicious for COVID-19 stated that the evaluation of CRP elevation together with biomarkers such as lactate dehydrogenase (LDH), ferritin, and D-dimer could predict the correct diagnosis with high specificity and sensitivity [33]. In the review published by Masoomikarimi M et al. on COVID-19 immunopathogenesis; they stated that besides CRP, there was an increase in pro-inflammatory and anti-inflammatory plasma cytokines such as IL-2, 4, 6, 7, 8, 10, IL-1β, IL-1RA, and tumor necrosis factor-alpha and they tried to shed on immunotherapy studies in

COVID-19 [34]. In our study, IL-34 was found to be successful in predicting the diagnosis of COVID-19. IL-34 was elevated in correlation with CRP in COVID-19-positive cases, and higher accuracy rates were also observed than CRP. Based on these results, studying IL-34 in people who apply to clinics with the suspicion of COVID-19 may be useful in supporting and predicting the diagnosis, and its use in clinical routine may be possible. In addition, as mentioned above, it has been understood that IL-34 plays a role in the immunopathogenesis of COVID-19 along with other cytokines, and this will guide further studies.

Although COVID-19 is a multi-system disease, one of its main target organs is the lung [35]. Both clinical and autopsy studies are continuing to understand the pathophysiology of lung involvement and the factors in patients' clinical deterioration. While lung involvement was thought to be primarily due to inflammatory and immune causes at the beginning of the pandemic, detailed studies on the pathophysiology of the disease showed that the situation is more complex than previously thought [35, 36]. In studies on the pathophysiology of lung involvement, mechanisms such as endothelial dysfunction, direct viral damage, intravascular diffuse coagulation and thrombo-embolism, pulmonary neo-angiogenesis, and pulmonary fibrosis have been implicated [35–37]. Smadja DM et al. reported in their study that there was a different clinical presentation from that seen in bacterial sepsis and disseminated intravascular coagulation (DIC) in severe COVID-19 patients [38]. Smadja DM et al and Norooznezhad AH et al. in studies where they evaluated the pathophysiology of COVID-19 SARS, endothelial dysfunction, pathological angiogenesis, pulmonary coagulation, and thromboembolism were stated as the main factors besides inflammatory processes [38, 39]. They reported that endothelial dysfunction occurs as a result of the SARS-CoV-2 spike protein interacting with an angiotensin-converting enzyme (ACE-2) and that ACE-2 expression increases in COVID-19 [38, 39]. In our study, IL-34 and CRP levels were higher in COVID-19-positive patients compared to the control group, but there was no difference in elevation between patients with and without lung involvement. On the other hand, there was a strong correlation between ferritin, D-dimer, and TnI and lung involvement. This result can be interpreted as the other aforementioned mechanisms may be at the forefront besides inflammatory lung involvement, in the light of the studies conducted on the pathophysiology of COVID-19 lung involvement mentioned above. Detailed studies are ongoing to solve the pathophysiology of lung involvement.

The limitations of our study can be stated as the relatively small number of cases and the fact that it was single-centered. In addition, other inflammatory and non-inflammatory markers mentioned above could not be studied in order to fully understand the pathophysiology of COVID-19 prognosis and lung involvement. However, it is valuable as it is one of the rare studies on IL-34 in COVID-19.

## Conclusion

As a result, IL-34 levels were found to be statistically significantly higher in COVID-19 patients compared to the control group and IL-34 was considered successful in predicting the diagnosis of COVID-19. However, IL-34 was not significant in predicting lung involvement and ICU admission. Our study has provided important outputs in determining the role of IL-34 in COVID-19 and should be supported by further studies.

## Supporting information

**S1 Data.**
(XLSX)

**S1 File.**
(PDF)

## Author Contributions

**Conceptualization:** Doğu Karahan, Hasan Ata Bolayir, Aslı Bolayir.

**Data curation:** Doğu Karahan, Hasan Ata Bolayir, Bilgehan Demir.

**Formal analysis:** Hasan Ata Bolayir, Önder Otlu, Mehmet Erdem.

**Investigation:** Doğu Karahan, Aslı Bolayir, Bilgehan Demir, Önder Otlu, Mehmet Erdem.

**Methodology:** Doğu Karahan, Aslı Bolayir, Önder Otlu, Mehmet Erdem.

**Project administration:** Doğu Karahan, Hasan Ata Bolayir.

**Resources:** Doğu Karahan, Bilgehan Demir, Önder Otlu, Mehmet Erdem.

**Software:** Hasan Ata Bolayir, Önder Otlu, Mehmet Erdem.

**Supervision:** Doğu Karahan, Hasan Ata Bolayir, Aslı Bolayir.

**Validation:** Doğu Karahan, Hasan Ata Bolayir, Aslı Bolayir, Önder Otlu, Mehmet Erdem.

**Visualization:** Hasan Ata Bolayir, Aslı Bolayir.

**Writing – original draft:** Doğu Karahan.

**Writing – review & editing:** Doğu Karahan, Hasan Ata Bolayir.

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
