## [Decision Letter · Decision Letter 0]

30 Jan 2024

PONE-D-23-19398CAN SERUM INTERLEUKIN 34 LEVELS BE USED AS AN INDICATOR FOR THE PREDICTION AND PROGNOSIS OF COVID-19?PLOS ONE

Dear Dr. karahan,

Thank you for submitting your manuscript to PLOS ONE. After careful consideration, we feel that it has merit but does not fully meet PLOS ONE’s publication criteria as it currently stands. Therefore, we invite you to submit a revised version of the manuscript that addresses the points raised during the review process.

We look forward to receiving your revised manuscript.

Kind regards,

Nilanka Perera, MD, PhD

Academic Editor

PLOS ONE

Journal Requirements:

https://www.mdpi.com/2075-4426/11/9/891/htm

https://academic.oup.com/jleukbio/article-abstract/104/5/931/6935404?redirectedFrom=fulltext&login=false

https://pubmed.ncbi.nlm.nih.gov/29313563/

In your revision ensure you cite all your sources (including your own works), and quote or rephrase any duplicated text outside the methods section. Further consideration is dependent on these concerns being addressed.

4. In the online submission form, you indicated that "The data is available from the author and will be provided if needed.". 

Additional Editor Comments:

**The sample size of 40 (20 in critically ill and non-severe categories) is not clear. Please justify how you arrived at the sample size to address the research questions.**

**Statistical analysis mention that data in a skewed distribution is given as median (max-min). Tables have median (IQR). Please correct the text.**

**Figure legends are not adequate. It should stand out. Please describe the figures appropriately. **

**Make sure that sentences do not start with a number. Please edit**

**References: Some have the doi while others do not. Add the doi for all references**

**Conclusion: One sentence on mortality due to thromboembolic phenomenon is not appropriate to the results of the study. Should conclude based on the current study results.**

**Describe the ICU group (ventilator req, NIV support...etc). **

Reviewers' comments:

Reviewer's Responses to Questions

**Comments to the Author**

1. Is the manuscript technically sound, and do the data support the conclusions?

Reviewer #1: Yes

Reviewer #2: Yes

2. Has the statistical analysis been performed appropriately and rigorously? 

Reviewer #1: Yes

Reviewer #2: Yes

3. Have the authors made all data underlying the findings in their manuscript fully available?

Reviewer #1: Yes

Reviewer #2: No

4. Is the manuscript presented in an intelligible fashion and written in standard English?

Reviewer #1: Yes

Reviewer #2: Yes

5. Review Comments to the Author

Reviewer #1: While the authors' attempt to find answers to dilemmas of clinical significance is greatly appreciable, there are a few things that warrant further clarification.

1. Firstly, have all the COVID-19-positive patients (excluding dexamethasone administration and age <18) mentioned in the said period between 01.09.2022 and 24.10.2022 included in the study?

2. Were patients whose IL-34 levels drawn before dexamethasone administration included in the study? How come the ICU patients included in the study not receive dexamethasone?

3. How was the control group selected? While the COVID-19-negative patients were included in the study, on what basis were the specific 40 patients selected?

4. Why were patients with CO-RADS 3 eliminated from the study?

5. Were the autoimmune or cancer history evaluated for the included patients to see if either is contributing to the elevated IL-34?

6. Was upper respiratory viral panel tested on all the patients? Was the possibility of concurrent viral infections causing elevated IL-34 evaluated?

7. Line 189-190 appears to be an incomplete sentence.

8. What is the relevance of Paragraph 4 i.e. Line 193-208 for this study?

As mentioned earlier, this is a great topic of relevance that the authors have targeted. However, the authors should provide concrete evidence to strongly support the study's conclusion eliminating all confounding factors.

Reviewer #2: In this study, IL-34 levels were studied in 40 COVID-19- positive patients and 40 COVID-19- negative patients to provide information about the disease's diagnosis, severity, and prognosis. They found that IL-34 was successful in predicting the diagnosis of COVID-19.

This is a very original work, but I have some objections and questions:

1- The number of patients studied is small.

2- The characteristics of ICU patients need to be clarified (such as their underlying diseases, disease severity, respiratory support treatments they receive, severity of lung involvement, given treatments other than steroids, etc.)

3- Why were patients tested for hs-CRP instead of CRP? In addition, why were lipid profile and troponin I examined in the patients? IL-34 has been associated with cardiovascular complications in some COVID studies (see references provided). Did the authors have such a purpose in this study?

Clinical Implications of IL-32, IL-34 and IL-37 in Atherosclerosis: Speculative Role in Cardiovascular Manifestations of COVID-19.

Law CC, Puranik R, Fan J, Fei J, Hambly BD, Bao S.Front Cardiovasc Med. 2021 Aug 6;8:630767. doi: 10.3389/fcvm.2021.630767.

Association of Interleukin-32 and Interleukin-34 with Cardiovascular Disease and Short-Term Mortality in COVID-19.

Kaufmann CC, Ahmed A, Muthspiel M, Rostocki I, Pogran E, Zweiker D, Burger AL, Jäger B, Aicher G, Spiel AO, Vafai-Tabrizi F, Gschwantler M, Fasching P, Wojta J, Huber K.J Clin Med. 2023 Jan 27;12(3):975. doi: 10.3390/jcm12030975.

4- The introduction and discussion part of the article are very long. Especially in the discussion section, between lines 193 and line 257 contain unnecessary details. If the information here is desired to be associated with the study results or the purpose of the study, more effort should be made to clarify.

5- Many parts of the article mentioned that IL-34 is an essential determinant in diagnosing COVID-19, and even the authors gave a cut-off value, saying that "IL-34 levels higher than 31.75 pg/mL can predict COVID-19". I have an objection to this finding and its interpretation.

COVID-19 is diagnosed by performing a PCR test on the taken samples to show the SARS-CoV-2 virus. IL-34 and other cytokines are shaped according to the immune response of the person (host) and determine the severity of the disease, organ involvement, and prognosis.

6. PLOS authors have the option to publish the peer review history of their article (what does this mean?). If published, this will include your full peer review and any attached files.

Reviewer #1: No

Reviewer #2: **Yes: **Gulbin Aygencel

---

## [Author Response · Author response to Decision Letter 0]

12 Feb 2024

Dear Editor,

 We thank you for your kind reply to our paper (PONE-D-23-19398) entitled " CAN SERUM INTERLEUKIN 34 LEVELS BE USED AS AN INDICATOR FOR THE PREDICTION AND PROGNOSIS OF COVID-19?’’. We have evaluated your precious comments on our manuscript and have revised our article in light of your suggestions. Below you can find our answer. We will be very pleased if you re-evaluate our work. 10.02.2024 

Kind regards.

Doğu Karahan

Corresponding author

dogu.karahan@ozal.edu.tr

1. The text was edited according to PLOS ONE style requirements.

2. The authors revised the manuscript in terms of written language. Complex sentences were tried to be made more understandable. Grammar errors corrected. However, if you still request correction after your evaluation, professional editing will be done.

3. The references were reviewed, and the publication in which you stated that there were similarities was also cited and added to the references. (Baghdadi M, Umeyama Y, Hama N, Kobayashi T, Han N, Wada H, et al. Interleukin-34, a comprehensive review. J Leukoc Biol. 2018 Nov;104(5):931-951. doi: 10.1002/JLB.MR1117-457R. )

4. The data set was added to the revision file as a supplementary file.

Additional Editor Comments:

1. The sample size of 40 (20 in critically ill and non-severe categories) is not clear. Please justify how you arrived at the sample size to address the research questions.

Our Answer: Power analysis was performed in terms of the sample size of the study and was added to the statistical analysis section with reference. Power analysis has been uploaded to the system as an additional document. Although the information you requested in terms of sample size is provided, the following reasons were effective in the relatively small number.

Reviewers are right the value of the study would increase if the number of cases were higher. But there were several reasons for the low number of cases. In fact, if it had been possible to start this study earlier, more number of cases could have been obtained. However, during the period when we conducted the study, there was a decrease in the number of patients in need of intensive care, a decrease in the intensity of the COVID-19 pandemic in our region, and a significant decrease in hospital admissions. The cases included in the study from the ICU were the cases who were admitted to the ICU within the first 24 hours after hospital admission. This was another factor in the relatively low number of ICU cases. Again, one of the main reasons was that the purchased kits were expensive. No funding support was received for this study. For non-routine examinations, we used kits we purchased with our resources.

2. Statistical analysis mention that data in a skewed distribution is given as median (max-min). Tables have median (IQR). Please correct the text.

Our answer: Inconsistencies between the tables and the spelling in the statistical analysis section have been corrected. We made adjustments to the tables to ensure harmony. 

 3. Figure legends are not adequate. It should stand out. Please describe the figures appropriately. 

Our answer: Figure legends have been re-arranged

 4. Make sure that sentences do not start with a number. Please edit

 Our answer: Sentences starting with numbers were corrected.

 5. References: Some have the doi while others do not. Add the doi for all references

 Our answer: Missing doi numbers of references have been added.

 6. Conclusion: One sentence on mortality due to thromboembolic phenomenon is not appropriate to the results of the study. Should conclude based on the current study results.

Our answer: Edits were made in the Conclusion section in line with your suggestions.

 7. Describe the ICU group (ventilator req, NIV support...etc). 

 Our answer: ICU group features were explained and added to the material method section.

Reviewer 1: 

Dear reviewer, thank you for your evaluation and suggestions. We have made adjustments in line with your suggestions. We will be very pleased if you re-evaluate our work.

Kind regards.

1. Firstly, have all the COVID-19-positive patients (excluding dexamethasone administration and age <18) mentioned in the said period between 01.09.2022 and 24.10.2022 included in the study?

Our answer: All patients who met the inclusion and exclusion criteria between 01.09.2022 and 24.10.2022 were tried to be included in the study. In fact, if it was possible to start this study earlier, more cases could be obtained. However, during this period, there was a decrease in the number of COVID-19 patients, especially in need of intensive care, and there was also a decrease in the intensity of the COVID-19 pandemic in our region.

2. Were patients whose IL-34 levels drawn before dexamethasone administration included in the study? How come the ICU patients included in the study not receive dexamethasone?

Our answer: IL-34 levels were measured before dexamethasone administration. The cases included in the study from ICU were the cases who were admitted to the intensive care unit within the first 24 hours after our COVID-19 clinical or emergency service admission. This was one of the factors in the relatively low number of cases.

3. How was the control group selected? While the COVID-19-negative patients were included in the study, on what basis were the specific 40 patients selected?

Our answer: The COVID-19 negative group consisted of cases with similar age and demographic structure as the COVID-19 positive group, who applied with complaints and symptoms of respiratory tract infection, and had negative PCR test results and normal lung imaging.

Power analysis was performed in terms of the sample size of the study and was added to the statistical analysis section with reference in determining the number of patients.

4. Why were patients with CO-RADS 3 eliminated from the study?

Our answer: As you know, the group classified as CO-RADS 3 is indicated in images that cannot be distinguished from COVID-19 and other viral, bacterial, rheumatological or other clinical conditions with lung involvement. As stated in the material and method section, IL-34, CRP, ferritin, D-dimer, and TnI were compared in terms of lung involvement in COVID-19-positive patients. For these reasons, CO-RADS 3 has been excluded to avoid confusion and criticism regarding other causes of lung involvement.

5. Were the autoimmune or cancer history evaluated for the included patients to see if either is contributing to the elevated IL-34?

Our answer: Chronic diseases were recorded by examining the patients' files and querying their CV data. Among the cases, there were no patients with a history of cancer or autoimmune rheumatological disease.

6. Was upper respiratory viral panel tested on all the patients? Was the possibility of concurrent viral infections causing elevated IL-34 evaluated?

Our answer: The period in which we conducted the study was the period when the COVID-19 pandemic was dominant, and the diagnoses of the cases were confirmed by COVID-19 PCR and thorax CT. During the pandemic, other viral agents were not routinely studied in our region, and the clinic where we worked did not have testing kits containing broad parameters for screening these respiratory factors. For these reasons, a separate evaluation could not be made in terms of other viral respiratory causes.

7. Line 189-190 appears to be an incomplete sentence

Our answer: Lines 189-190 were edited and tried to be more clearly understandable.

8. What is the relevance of Paragraph 4 i.e. Line 193-208 for this study?

Our answer: The purpose of this paragraph was to discuss the fact that there was no correlation between IL-34 level and patient prognosis, mortality risk, lung involvement, and intensive care admission risks and that the situation underlying these prognostic features may be related to thromboembolic causes rather than inflammatory processes. Again, in our study, this thesis was discussed and supported through D-dimer and Tn I, which were found to be statistically significant in terms of prognosis, mortality risk, lung involvement, and intensive care hospitalization risks. However, as you stated, this paragraph was considered to be too long and detailed, and therefore, edits were made in this paragraph and shortened.

Reviewer 2: 

Dear reviewer, thank you for your evaluation and suggestions. We have made adjustments in line with your suggestions. We will be very pleased if you re-evaluate our work.

Kind regards.

1.The number of patients studied is small.

Our answer: You are right that the value of the study would increase if the number of cases were higher. But there were several reasons for the low number of cases. In fact, if it had been possible to start this study earlier, more number of cases could have been obtained. However, during the period when we conducted the study, there was a decrease in the number of patients in need of intensive care, a decrease in the intensity of the COVID-19 pandemic in our region, and a significant decrease in hospital admissions. The cases included in the study from ICU were the cases who were admitted to ICU within the first 24 hours after hospital admission. This was another factor in the relatively low number of ICU cases. Again, one of the main reasons was that the purchased kits were expensive. No funding support was received for this study. For non-routine examinations, we used kits we purchased with our resources.

2- The characteristics of ICU patients need to be clarified (such as their underlying diseases, disease severity, respiratory support treatments they receive, severity of lung involvement, given treatments other than steroids, etc.)

Our answer: Characteristics of ICU patients and treatment information were added to the manuscript.

3. Why were patients tested for hs-CRP instead of CRP? In addition, why were lipid profile and troponin I examined in the patients? IL-34 has been associated with cardiovascular complications in some COVID studies (see references provided). Did the authors have such a purpose in this study?

Our answer: Following your warning about hs-CRP, we contacted to the laboratory again and confirmed that the study was carried out as CRP, not hs-CRP. The involuntary mistake has been corrected. We made the necessary corrections regarding CRP in the main text.

As you know very well, the factors that play a role in the course of COVID-19 and the increase in mortality risk have not been fully elucidated. COVID-19 continues to have the feature of a multi-unknown equation. There are many studies indicating that thrombo-embolic events and cardiovascular events play an important role in the course of COVID-19. Considering the predictions that COVID-19 increases the risk of coagulopathy, the troponın I, D-dimer and cholesterol profile was also examined and evaluated. And again, the possible relationship between COVID-19 and thrombo embolism is discussed in the discussion section. The reference you suggested regarding the relationship between IL-34 and COVID-19 and thromboembolism has also been evaluated and added to the discussion. (Kaufmann CC, Ahmed A, Muthspiel M, Rostocki I, Pogran E, Zweiker D, et al. Association of Interleukin-32 and Interleukin-34 with Cardiovascular Disease and Short-Term Mortality in COVID-19. J Clin Med. 2023 Jan 27;12(3):975. doi: 10.3390/jcm12030975.)

Again, when the referenced study you suggested was evaluated, it was a review evaluating the relationship between cardiovascular diseases and IL-34, and based on this, it was speculated that IL-34 may be related to increased cardio-thromboembolic risks in COVID-19. The study did not provide any research or evidence on IL-34 and COVID-19. Since it reached conclusions based on interpretation, it was not included in the article. (Clinical Implications of IL-32, IL-34 and IL-37 in Atherosclerosis: Speculative Role in Cardiovascular Manifestations of COVID-19. Law CC, Puranik R, Fan J, Fei J, Hambly BD, Bao S.Front Cardiovasc Med. 2021 Aug 6;8:630767. doi: 10.3389/fcvm.2021.630767.) 

4. The introduction and discussion part of the article are very long. Especially in the discussion section, between lines 193 and line 257 contain unnecessary details. If the information here is desired to be associated with the study results or the purpose of the study, more effort should be made to clarify.

Our answer: As you suggested, the unnecessary details you mentioned, especially in the discussion section were reviewed and the long and detailed sections were edited and shortened.

5. Many parts of the article mentioned that IL-34 is an essential determinant in diagnosing COVID-19, and even the authors gave a cut-off value, saying that "IL-34 levels higher than 31.75 pg/mL can predict COVID-19". I have an objection to this finding and its interpretation.

COVID-19 is diagnosed by performing a PCR test on the taken samples to show the SARS-CoV-2 virus. IL-34 and other cytokines are shaped according to the immune response of the person (host) and determine the severity of the disease, organ involvement, and prognosis.

Our answer: You are absolutely right that the main way to confirm the diagnosis of COVID-19 is to demonstrate the virus by PCR. There are not many studies conducted with IL-34 on COVID-19. Based on this, we wanted to study the importance of IL-34. You are right that we cannot diagnose COVID-19 with IL-34 alone, and IL-34 is not a routine test. Additionally, it may currently be considered an expensive test that would be difficult to use routinely. In fact, as we stated in the paragraph where we discussed IL-34 from this perspective, parameters such as CRP, LDH, ferritin, and D-dimer are not used in the definitive diagnosis of COVID-19, but have been studied in previous studies from a predictive perspective. We determined the cut-off value in order to support future studies and to provide guidance in a situation where the routine use of IL-34 may come to the fore and to bring COVID-19 to mind in the differential diagnosis. Again, as you mentioned, we focused on evaluating the host's immune response and prognosis through IL-34.

---

## [Decision Letter · Decision Letter 1]

19 Mar 2024

PONE-D-23-19398R1Can serum interleukin 34 levels be used as an indicator for the prediction and prognosis of COVID-19?PLOS ONE

Dear Dr. karahan,

Thank you for revising  your manuscript and addressing majority of reviewer comments. As there is a minor suggestion to improve the paper, we suggest you to edit and submit the revision to consider for publication.  

We look forward to receiving your revised manuscript.

Kind regards,

Nilanka Perera, MD, PhD

Academic Editor

PLOS ONE

Journal Requirements:

Reviewers' comments:

Reviewer's Responses to Questions

**Comments to the Author**

1. If the authors have adequately addressed your comments raised in a previous round of review and you feel that this manuscript is now acceptable for publication, you may indicate that here to bypass the “Comments to the Author” section, enter your conflict of interest statement in the “Confidential to Editor” section, and submit your "Accept" recommendation.

Reviewer #1: All comments have been addressed

Reviewer #2: All comments have been addressed

2. Is the manuscript technically sound, and do the data support the conclusions?

Reviewer #1: Yes

Reviewer #2: Yes

3. Has the statistical analysis been performed appropriately and rigorously? 

Reviewer #1: Yes

Reviewer #2: Yes

4. Have the authors made all data underlying the findings in their manuscript fully available?

Reviewer #1: Yes

Reviewer #2: Yes

5. Is the manuscript presented in an intelligible fashion and written in standard English?

Reviewer #1: Yes

Reviewer #2: Yes

6. Review Comments to the Author

Reviewer #1: I appreciate the authors clarifying the questions posed. Couple of unanswered questions.

1. How come the ICU patients included in the study did not receive

dexamethasone?

2. Were the autoimmune or cancer history evaluated for the included patients to see if either is contributing to the elevated IL-34? While this question was partly clarified, did any of the patients get serologic testing for antibodies? Or were assumed to be negative based on clinical history. Especially with COVID having a lot of respiratory and renal failure similar to a lot of pulmonary/renal syndromes associated with vasculitis, were they definitely ruled out?

3. I would recommend some of the explanations here be incorporated into the article for better clarity.

Reviewer #2: (No Response)

7. PLOS authors have the option to publish the peer review history of their article (what does this mean?). If published, this will include your full peer review and any attached files.

Reviewer #1: No

Reviewer #2: No

---

## [Author Response · Author response to Decision Letter 1]

21 Mar 2024

Dear Editor,

 We thank you for your kind reply to our paper (PONE-D-23-19398) entitled " CAN SERUM INTERLEUKIN 34 LEVELS BE USED AS AN INDICATOR FOR THE PREDICTION AND PROGNOSIS OF COVID-19?’’. We have evaluated your precious comments on our manuscript and have revised our article in light of your suggestions. Below you can find our answer. We will be very pleased if you re-evaluate our work. 21.03.2024 

Kind regards.

Doğu Karahan

Corresponding author

dogu.karahan@ozal.edu.tr

Journal Requirements:

Our answer: References have been checked. The doi number of reference number 1 has been added. No additional abnormalities were observed in the references.

Reviewer 1: 

Dear reviewer, thank you for your evaluation and suggestions. We have made adjustments in line with your suggestions. We will be very pleased if you re-evaluate our work.

Kind regards.

1. How come the ICU patients included in the study did not receive dexamethasone?

Our answer: Dexamethasone treatment was administered to intensive care COVID-19 patients. However, IL-34 levels were measured before the first dexamethasone administration. The cases included in the study from ICU were the cases who were admitted to the intensive care unit within the first 24 hours after our COVID-19 clinical or emergency service admission. After blood samples were taken, dexamethasone treatments were administered to the patients. When we reviewed the article again, we realized that we could not explain this part clearly, and we added it more clearly to the material and method section as you suggested.

2. Were the autoimmune or cancer history evaluated for the included patients to see if either is contributing to the elevated IL-34? While this question was partly clarified, did any of the patients get serologic testing for antibodies? Or were assumed to be negative based on clinical history. Especially with COVID having a lot of respiratory and renal failure similar to a lot of pulmonary/renal syndromes associated with vasculitis, were they definitely ruled out?

Our answer: It was decided that the cases were not diagnosed as auto-immune or cancer, based on their medical history and clinical status. Since there was no clinical suspicion in the follow-up of the cases, serological testing or further examination and evaluation were not performed in these respects. You are right that there could be many overlaps with COVID-19, similar to vasculitic lung/kidney syndrome. No detailed evaluation has been made on this issue. If you wish, we can add this situation to the limitations of the study section.

3. I would recommend some of the explanations here be incorporated into the article for better clarity.

Our answer: When we reviewed the article again, we realized that we could not clearly explain the parts you mentioned above, and we added it more clearly to the material and method section as you suggested.

---

## [Editor Report · Decision Letter 2]

26 Mar 2024

Can serum interleukin 34 levels be used as an indicator for the prediction and prognosis of COVID-19?

PONE-D-23-19398R2

Dear Dr. karahan,

We’re pleased to inform you that your manuscript has been judged scientifically suitable for publication and will be formally accepted for publication once it meets all outstanding technical requirements.

Kind regards,

Nilanka Perera, MD, PhD

Academic Editor

PLOS ONE
---

## [Editor Report · Acceptance letter]

1 Apr 2024

PONE-D-23-19398R2 

PLOS ONE

Dear Dr. Karahan, 

I'm pleased to inform you that your manuscript has been deemed suitable for publication in PLOS ONE. Congratulations! Your manuscript is now being handed over to our production team.

Kind regards, 

on behalf of

Dr. Nilanka Perera 

Academic Editor

PLOS ONE